

# Assessment of the effects of transforming growth factor beta1 (TGF-β1)-Smad2/3 on fibrosis in rat myofascial trigger points using point shear wave elastography

Xin Fang[1,*], Yalong Yin[2,*], Haimei Lun[3], Shitao Su[1] and Shangyong Zhu[1]

[1] Department of Medical Ultrasound, First Affiliated Hospital of Guangxi Medical University, Nanning, Guangxi, China

[2] Department of Traditional Chinese Medicine, First Affiliated Hospital of Guangxi Medical University, Nanning, Guangxi, China

[3] Department of Ultrasound, People's Hospital of Guangxi Zhuang Autonomous Region, Nanning, Guangxi, China

* These authors contributed equally to this work.

Corresponding author
Shangyong Zhu,
zhushangyong2022@sina.cn

## ABSTRACT

**Background & Aims:** Myofascial trigger points (MTrPs) are highly sensitive irritated points within a tense belt of skeletal muscle, and are the main cause of muscle pain and dysfunction. MTrPs can also cause paraesthesia and autonomic nervous dysfunction. Furthermore, long-term and chronic MTrPs can cause muscle atrophy and even disability, seriously affecting the quality of life and mental health of patients, and increasing the social and economic burden. However, to date, there have been few studies on fibrogenesis and changes in MTrPs. Therefore, this study investigated whether transforming growth factor beta1 (TGF-β1)-Smad2/3 participates in the formation of MTrPs and how it affects fibrosis using point shear wave elastography.

**Methods:** Forty Sprague–Dawley rats were randomly divided into the MTrPs group and the control group. Blunt injury combined with eccentric exercise was used to establish an MTrPs model. Electromyography (EMG), haematoxylin and eosin (H&E) staining and transmission electron microscopy (TEM) were used to verify the model. The collagen volume fraction was measured by Masson staining, the protein expression of TGF-β1 and p-Smad2/3 was measured by Western blotting (WB) and immunohistochemistry (IHC), and the shear wave velocity (SWV) was measured by point shear wave elastography.

**Results:** EMG, H&E and TEM examination indicated that the modelling was successful. The collagen volume fraction and the protein expression of TGF-β1 and p-Smad2/3 were higher in the MTrPs group than in the control group. The SWV of the MTrPs group was also higher than that of the control group. These differences suggest that MTrPs may exhibit fibrosis. The correlations between the collagen volume fraction and SWV and between the collagen volume fraction and TGF-β1 were positive.

**Conclusion:** Fibrotic conditions may be involved in the formation of MTrPs. Ultrasound point shear wave elastography and assessment of TGF-β1 and p-Smad2/3 expression can reflect the degree of MTrPs fibrosis to some extent. Further

exploration of the important role of TGF-β1 and Smad2/3 in the pathogenesis of MTrPs will be of great significance for clinical treatment.

## INTRODUCTION

Myofascial trigger points (MTrPs) can be associated with a reduced range of motion through muscle dysfunction and muscle weakness (*Vadasz et al., 2020*). Athletes who are undergoing preseason training can also suffer muscle overload injuries and develop trigger points. There are several theories about how trigger points develop. One accepted theory is that trigger points are the result of muscle injury, overuse, and spasm (*Money, 2017*). Histopathological and electrophysiological studies have shown that chronic MTrPs are chronic muscle lesions caused by local energy supply disorders secondary to acute and chronic injury. Analysis of electrophysiological signals has revealed the presence of spontaneous electrical activity (SEA) at MTrPs (*Ge, Fernández-de-Las-Peñas & Yue, 2011*) and confirmed that the SEA is due to excessive release of acetylcholine from an abnormal endplate. Histopathologically, MTrPs manifest as a tense belt formed by contraction of muscle fibres. However, the current diagnosis of MTrPs relies on clinical palpation, which is highly dependent on the skill and judgement of the examiner. Unfortunately, there is a lack of objective and quantitative diagnostic measures for tense belts. Therefore, there is a high demand for imaging techniques that provide quantitative and objective measurements for tense belts and MTrPs. Magnetic resonance imaging (MRI) and ultrasound (US) have been used to identify MTrPs for diagnosis and treatment evaluation (*Jiang et al., 2020*; *Liang, Guo & Li, 2021*; *Sikdar et al., 2009*; *Yu et al., 2021*). Although both are noninvasive examination techniques, US has faster data acquisition and a lower cost than MRI. Point shear wave elastography involves the use of a set of shear waves to induce normal and directional tissue displacement at a single focal point. The velocity of the shear waves vertical to the plane of induction is calculated, and higher shear wave speeds and smaller displacements are associated with stiffer tissues (*Nightingale, 2011*).

After tissue damage, fibrosis and antifibrotic factors influence each other, forming a network. Transforming growth factor beta1 (TGF-β1) is an important profibrotic factor. Research has shown that it plays a key role in fibrosis of the lungs, kidneys, liver, skin and skeletal muscle and is the core factor in the fibrotic process (*Ismaeel et al., 2019*). TGF-β1 promotes the formation and contraction of the extracellular matrix (ECM) as well as the transformation of epithelial cells into fibroblasts. It is present in the damaged area throughout the repair process. Smad proteins are key molecules in the signal transduction process of TGF-β family members, and they participate in most cellular and physiological functions of the TGF-β superfamily. In addition, many researchers have found that the TGFβ-Smad signalling pathway is closely related to fibrosis in a variety of other tissues. *Xiao et al. (2017)* found that TGF-β1 and Smad2/3 levels are significantly increased in
myocardial infarction scar tissue. Increased expression of Smad7 inhibits the expression of collagen type I and αSMA and alleviates skin scar hyperplasia (*Kopp et al., 2005*). *Goto et al. (2004)* found that an increase in Smad4 expression can promote the occurrence of glomerulonephritis and fibrosis in mice.

In the present study, a rat MTrPs model based on blunt injury combined with eccentric movement was established as described in previous studies (*Huang et al., 2013*). The aim was to explore whether fibrotic conditions may be involved in the formation of MTrPs. The correlations between the collagen volume fraction and SWV and the collagen volume fraction and the expression of TGF-β1 and phosphorylated(p-)Smad2/3 were also investigated.

# MATERIALS AND METHODS

## Experimental animals and grouping

Forty 5-week-old specific pathogen-free Sprague−Dawley rats (healthy males) were purchased from the Animal Experimental Center of Guangxi Medical University (Certificate of Conformity: SCXK Gui2009-0002) and randomly divided into a control group ($n = 10$) and an MTrPs group ($n = 30$).The groups of animals were housed separately at a density of three animals per cage. The environment was controlled, with a humidity of 50–70%, a temperature of 22–24 °C, and a 12/12 h light/dark cycle.

## MTrPs model rats

There is general agreement that muscle overuse or direct trauma to the muscle can lead to the development of MTrPs (*Money, 2017*). The MTrPs model was established as described by *Huang et al. (2013)* *via* a blunt strike in combination with eccentric exercise (Fig. 1) for 8 weeks. In this study, the right vastus medialis muscle was selected as the modelling area, and the remaining procedure was the same as that described in the reference literature. The model was considered successful on the basis of the following criteria: the right vastus medialis muscle exhibited one or more tense belts and contracture nodules on palpation, a local twitch response (LTR) was produced by insertion of an electrode needle into the nodules, high-frequency spontaneous electrical activity (SEA) was observed on electromyography (EMG) under resting conditions, and a continuous spontaneous potential occurred for more than 30 s. The region was marked when two or more of the above criteria were consistent with MTrPs.

## Paw withdrawal threshold assessment

The paw withdrawal threshold was measured with von Frey filaments (*Nahm et al., 2017*) at 2, 4, 6 and 8 weeks of modelling and at 2 and 4 weeks after the completion of modelling. The weight of each animal was measured between 13:00 and 18:00 PM. The temperature of the test environment was controlled at between 24 °C and 26 °C, the humidity was maintained between 50% and 70%, and the environment was kept quiet. Increasing stimulation pressure was manually applied with von Frey filaments of increasing stiffness (2, 4, 6, 8, 10, 15, 26 g) in the centre of the ball of the right foot in the rats. The filament stiffness was recorded when evidence of the paw withdrawal reflex occurred, such as lifting,

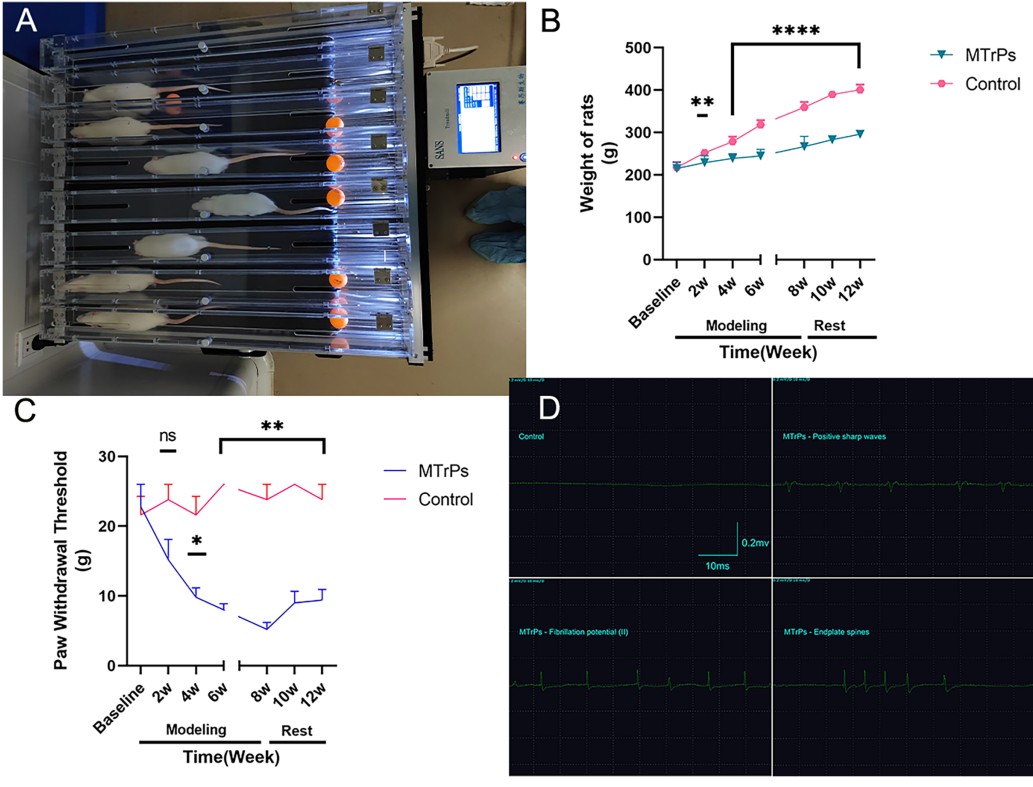

**Figure 1  General information.** (A) Eccentric exercise. (B) The weight gain of the MTrPs rats was slow, and there was a difference between the MTrPs group and the control group beginning in the second week. The weight gain of the MTrPs rats was faster during the rest stage than that of the model rats, but it was still significantly different from that of the control group. (C) In the MTrPs group, the paw withdrawal threshold (PWT) decreased gradually at the beginning of the modelling stage, and the PWT was markedly different from that of the control group beginning in the fourth week. Although the PWT of the MTrPs rats during the rest stage tended to be greater than that during the modelling stage, it was still significantly different from that of the control rats, and this difference lasted until the end of the experiment. (D) Results of the EMG examination. Compared to the control group, $^*p < 0.05$, $^{**}p < 0.01$, $^{****}p < 0.0001$, ns $p > 0.05$. The data were analysed by independent-sample T tests or Mann–Whitney U tests; repetitions = 3.

withdrawing, swinging or licking. The test was repeated five times for each rat with an interval of at least 2 min. The maximum and minimum values were removed, and the average of three tests was calculated as the paw withdrawal threshold (g).

## Electromyographic examination

Electromyography (EMG) was performed after the modelling was completed. All rats were intraperitoneally anaesthetized with sodium pentobarbital (50 mg/kg of body weight) and subjected to hair removal. The right lower limb region of each rat was completely exposed. The rats were then attached to a retainer. EMG equipment (Keypoint9033A07, Dendi EMG, Denmark) was used at a sampling frequency of 20 HZ–10,000 Hz, sensitivity of 0.2 mv/grid, and scanning speed of 10 ms/grid. Then, electroacupuncture electrodes (Φ0.3 mm, 30 gauge) were inserted into the significantly strained muscle belts to detect the LTR and SEA, including fibrillation potential (I/II), positive sharp waves, and endplate

spines. When the above SEA appeared with specific rhythmic waves on the electromyograph, images were recorded for 2 min. The rats in the control group were evaluated by EMG at the same position. All the above procedures were performed by an EMG physician with 8 years of experience.

## Ultrasound image processing

After the EMG examination, all rats were then subjected to ultrasound using the Mindray ResonaI9 Clinical Ultrasound (US) system (Mindray Biomedical Electronics Co., Ltd, Shenzhen, China) and a linear array transducer (L9-3 s) probe (3.6–9.0 MHz). The probe was gently placed on the marks for scanning, and the MTrPs in the right vastus medialis muscle were observed in multiple sections to avoid anisotropy artefacts. Then, using point shear wave elastography to measure the shear wave velocity (SWV) of MTrPs, a range of 3 mm × 3 mm was selected as the region of interest (ROI). The average value of three measurements was calculated for statistical analysis. The data are expressed in m/s. The same location was selected and measured in the control group. The above procedures were performed by two diagnostic sonographers with 8 and 10 years of experience.

## Haematoxylin and eosin (H&E) staining procedure

All rats were euthanized after point shear wave elastography examination and given an overdose of chloral hydrate, and the MTrPs tissues were perfused with paraformaldehyde (4%) through the heart for histological examination. The muscle with MTrPs was removed and fixed with 4% paraformaldehyde overnight. The tissues were subsequently embedded in paraffin and sectioned into 4 μm thick sections. Next, the sections were deparaffinized in xylene and rehydrated through a graded alcohol solution. Finally, the sections were stained with H&E and observed under a microscope at different magnifications (Eclipse Ci-L, Nikon, Toyko, Japan).

## Masson staining procedure

The MTrPs tissue sections were dewaxed and dehydrated, stained with haematoxylin for 5 min, washed with water and subjected to counterstaining for 5 min. The sections were stained with acid fuchsin solution for 5 min, washed with weak acid working solution for 1 min, and washed with phosphomolybdic acid solution for 1 min. The samples were then subjected to aniline blue staining for 1 min, washing with weak acid working solution for 1 min, gradient ethanol dehydration, and clearing, followed by sealing with neutral gum. The relationship between the blue collagen-positive area and the total tissue area was evaluated by microscopic observation. The collagen volume fraction was analysed and calculated with Aipathwell image analysis software (collagen volume fraction = collagen area in the visual field/measured visual field area).

## Transmission electron microscopy

MTrPs tissue blocks 1 mm * 2 mm * 3 mm in size were fixed in 2.5% glutaraldehyde phosphate buffer for 2 h. The samples were fixed, dehydrated, soaked, embedded, and cured according to typical electron microscopy preparation procedures. Thin slices subjected to 3% acetic acid uranium and lead nitrate staining were observed by

transmission electron microscopy (TEM) (HT7800/HT7700, Hitachi, Tokyo, Japan) and images of 30 randomly selected fields of view were captured.

### Immunohistochemistry

MTrPs tissues from each group were sliced at a thickness of 4 mm. In turn, the slices were placed into dewaxing solution, dehydrated in an alcohol gradient, and incubated in 3% hydrogen peroxide solution for 25 min at room temperature while protected from light. The tissues were blocked by dropping 3% BSA solution onto the sample for 30 min and incubated with rabbit anti-TGF-β1 (1:200, WL02193, Wanleibio, China), and rabbit anti-P-SMAD2/P-SMAD3 (1:200, WL02305, Wanleibio, China) primary antibodies at 4 °C overnight. The tissue was then covered with a secondary antibody (HRP labelled) of the same species as the primary antibody. DAB was used for colour development, the nuclei were restained, and the tissues were dehydrated for sealing and microscopic examination. The nuclei stained with haematoxylin were blue, and positive expression of DAB was indicated by light yellow to tan staining.

### Western blotting

MTrPs tissues were collected and lysed in RIPA buffer (Solarbio, Beiijng, China). Afterwards, a BCA protein assay kit was used to quantify the protein. Western blotting (WB) was carried out as previously described (*Lv et al., 2020*) with the following antibodies: rabbit anti-TGF-β1 (1:1,000,WL02193;Wanleibio, Beijing, China), and rabbit anti-P-SMAD2/P-SMAD3 (1:1,000, WL02305, Wanleibio, Beijing, China). Adobe Photoshop CS2 was used for target protein band cutting, and Image-Pro Plus was used to analyse the grey values of the protein bands.

### Statistical analysis

SPSS version 23.0 (Chicago, IL, USA) was used for data analysis, and GraphPad Prism (GraphPad Software, San Diego, CA, USA) was used to plot the data. All data are expressed as the mean (95% CI of the mean) or median (IQR). An independent sample T test was used to analyse the differences between groups, and the Mann–Whitney U test was used for nonnormally distributed variables. The correlations were determined by calculating Pearson linear correlation coefficients or Spearman rank correlation coefficients. Statistical significance was accepted at $p < 0.05$.

## RESULTS

### Animal model results

Of the 30 MTrPs rats, 15 rats were excluded from the experiment due to death from anaesthesia overdose, fracture, or substandard palpation and EMG examination. Therefore, 25 rats were included in the final analysis (MTrPs group: $n = 15$, control group: $n = 10$). A detailed description of the modelling is provided in Table 1.

### EMG results

The MTrPs were identified with EMG after 4 weeks of rest. Only a horizontal baseline appeared in the control group, while there was more SEA, as indicated by fibrillation

**Table 1 MTrPs modelling.**

| Groups | Substandard palpation and EMG | Anaesthesia overdose | Fracture | Final quantity |
|---|---|---|---|---|
| MTrPs ($n = 30$) | 10 | 3 | 2 | 15 |
| Control ($n = 10$) | 0 | 0 | 0 | 10 |

Note:
MTrPs, myofascial trigger points; EMG, electromyography.

**Table 2 Spontaneous electrical activity (SEA).**

| Group\SEA | Positive sharp waves | Fibrillation potential (II) | Endplate spines |
|---|---|---|---|
| MTrPs (n = 15) | 2 | 5 | 8 |
| Control (n = 10) | 0 | 0 | 0 |

potential (II) and the presence of positive sharp waves in the MTrPs group; in addition, increased numbers of endplate spines were observed in the MTrPs group (Fig. 1D, Table 2).

## Ultrasound STQ findings

Two-dimensional ultrasound examination of the MTrPs group (Figs. 2A and 2C) revealed abnormalities in or near tense belt regions. The main manifestations included uneven echo or low echo, increased blood flow and greater SWV. In the control group (Figs. 2B and 2D), there were no abnormalities on ultrasonography, only muscle fibres with a uniform echo and no blood flow.

## Comparison of pathological morphology and collagen volume fraction between the control group and the MTrPs group

**H&E staining.** In the MTrPs group, muscle cell disarrangement, fibrous connective tissue hyperplasia, neovascularization, inflammatory cell infiltration and nuclei in the interior or centre of the cells were observed (Fig. 3A). In the control group, the muscle cells were closely arranged, uniformly coloured, and clearly demarcated. Nuclei were located at the edge, and several muscle cells were clustered into bundles (Fig. 3B).

**Masson staining.** In the MTrPs group, red muscle fibres were disordered and squeezed, collagen fibre deposition increased significantly, and the fibres were thick and wrapped around new muscle cells and intermuscular membranes (Fig. 3C, yellow arrow). In the control group (Fig. 3D), the muscle fibres were arranged neatly, and the ratio of muscle fibres to collagen fibres was moderate.

## Comparison of skeletal muscle ultrastructure between the control group and the MTrPs group

Compared with the control group, the MTrPs group exhibited a disordered arrangement of muscle fibres, shorter sarcomeres ($P < 0.05$) (Fig. 4D),and a twisted and broken Z-line. The number of mitochondria was decreased ($P < 0.0001$) (Fig. 4E), the double-membrane

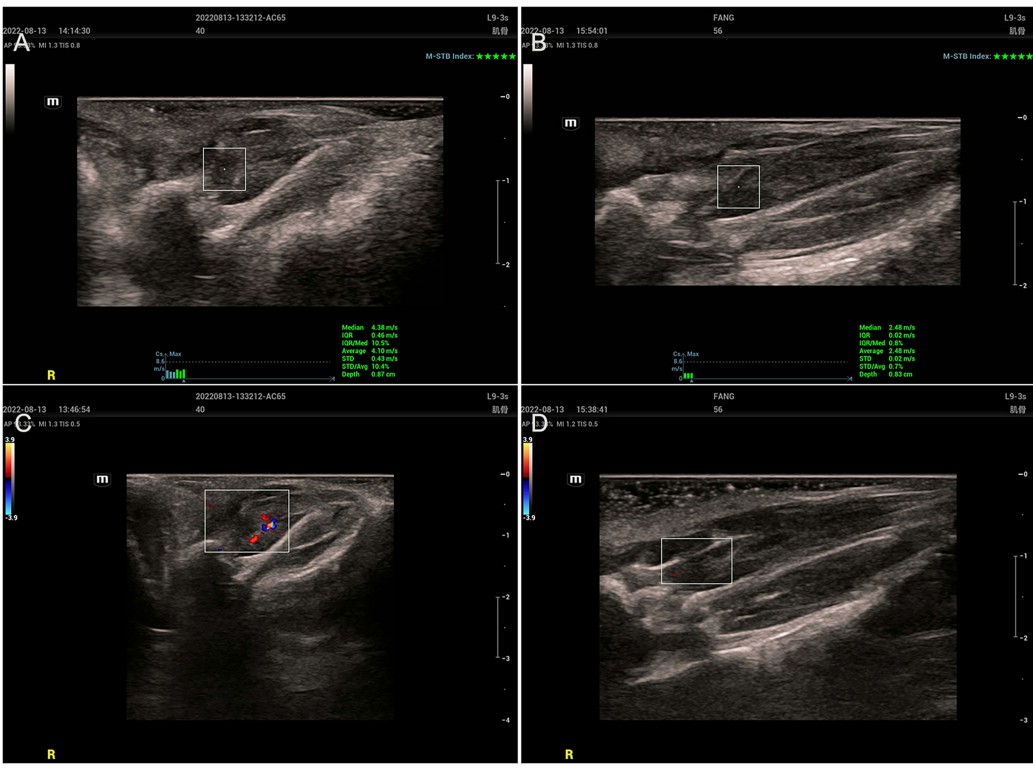

**Figure 2 Ultrasound findings.** (A and C) The MTrPs group showed richer blood flow and greater stiffness than the control group (B and D).

structure was blurred, and the sarcoplasmic reticulum was blurred in the MTrPs group. In the control group (Fig. 4C), the fibres of the vastus medialis muscle had an orderly arrangement, the myofilaments were evenly distributed, the length of the sarcomeres was uniform, and the Z line was clear and continuous. Mitochondria were evenly distributed among the myofibrils near the Z line, in an oval or long strip shape, with a normal size, morphology and structure and a visible bilayer membrane structure.

## Comparison of TGF-β1 and p-Smad2/3 expression between the control group and the MTrPs group

TGF-β1 and p-Smad2/3 in tissue were yellow-brown. The mean optical densities of TGF-β1 and p-Smad2/3 in the MTrPs group (0.011 ± 0.002 and 0.010 ± 0.002, respectively) were higher than those in the control group (0.009 ± 0.001 and 0.009 ± 0.001, respectively) ($p < 0.05$) (Figs. 5F and 5G).

## In the MTrPs group, the collagen volume fraction was linearly correlated with SWV and TGF-β1 (IOD/area)

For the MTrPs rats, the SWV was 3.61 ± 0.33 m/s, which was higher than that of the control rats (2.37 ± 0.20 m/s) ($p < 0.0001$; MTrPs group: $n = 15$, control group: $n = 10$) (Fig. 6A). No significant correlation was observed between SWV and rat weight/paw withdrawal threshold ($p > 0.05$) (Figs. 6B and 6C). In the MTrPs group, the collagen

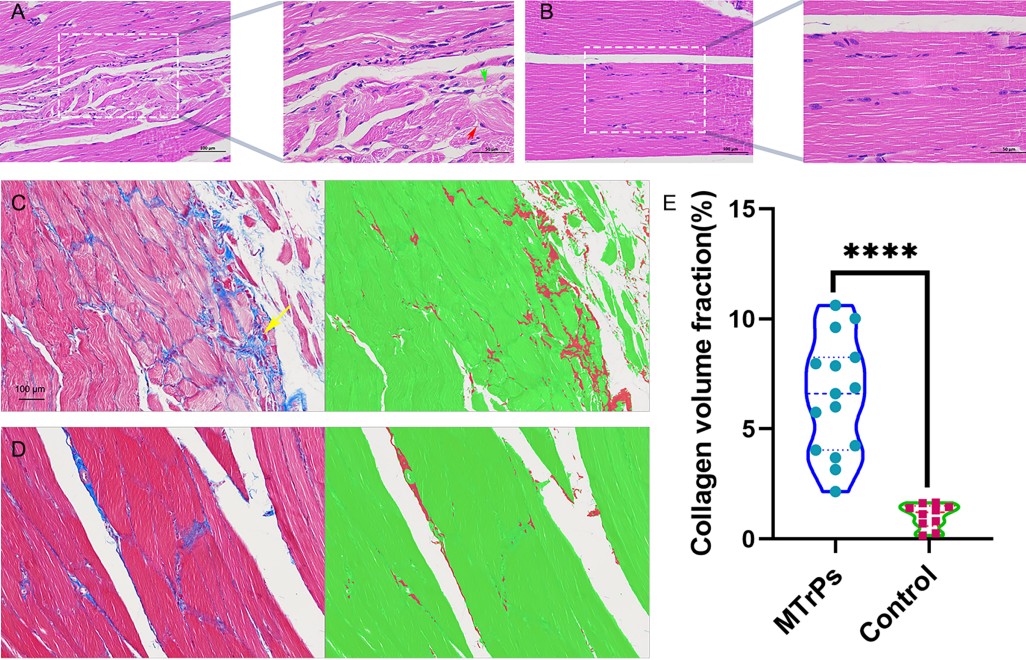

**Figure 3 Histopathological findings for the groups.** (A) Red arrow represent muscle cells with a disordered arrangement; green arrow represent fibrous connective tissue hyperplasia (H&E staining magnification: × 400/200, bar = 50/100 μm). (E) An increase in the collagen volume fraction indicates that MTrPs may undergo fibrosis. ****$p < 0.0001$, compared to the control group. The data were analysed by independent-sample T tests; repetitions = 3.

volume fraction was linearly correlated with SWV. The correlation coefficient between the collagen volume fraction and SWV was 0.539 ($p < 0.05$) (Fig. 6D). The correlation coefficient between the collagen volume fraction and TGF-β1 expression was 0.661 ($p < 0.01$) (Fig. 6E). No significant correlation was observed between the collagen volume fraction and p-Smad2/3 expression ($p > 0.05$) (Fig. 6F).

## DISCUSSION

An MTrPs rat model group and a normal control group were established in the present study.

EMG, H&E and TEM examination indicated that the MTrPs rat model was successfully established. The MTrPs group showed richer blood flow and greater stiffness than the control group, and an increase in the collagen volume fraction indicated that MTrPs may cause fibrosis. In addition, the protein expression of TGF-β1 and p-Smad2/3 was higher in the MTrPs group than in the control group. Finally, the correlations between the collagen volume fraction and SWV and between the collagen volume fraction and mean optical density of TGF-β1 were positive.

Despite the high prevalence of MTrPs (*Vadasz et al., 2020*), their diagnostic criteria are still unclear. Ultrasound studies have long been conducted to investigate the stiffness of MTrPs (*Sikdar et al., 2009*). In this study, in the MTrPs group rats, the SWV of the vastus medialis muscle was 3.61 ± 0.33 m/s, which was higher than that of the control group (Fig. 6). These results are consistent with those of previous studies on stiffness (*Jiang et al.,*

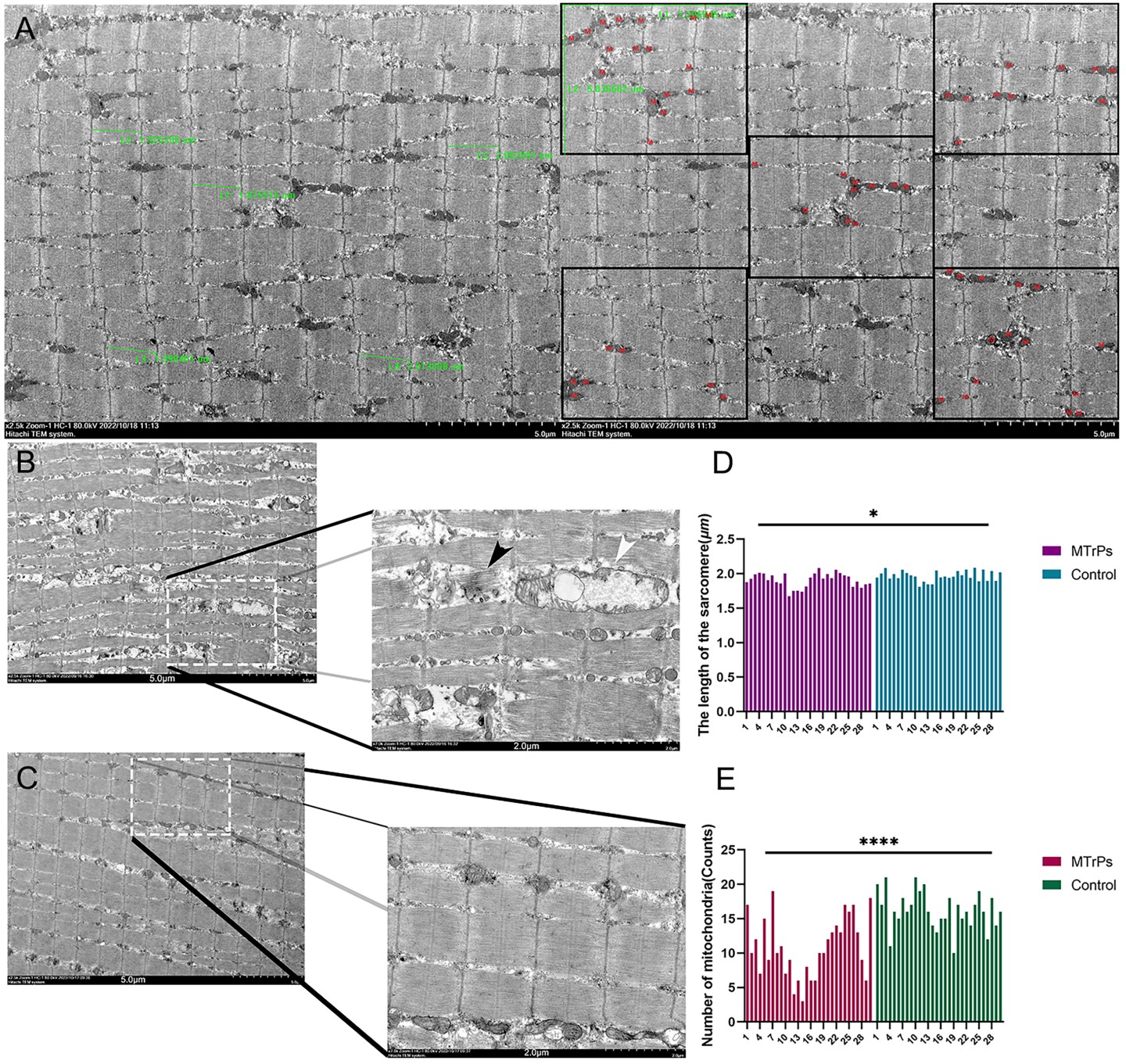

**Figure 4 Comparison of sarcomere length and the number of mitochondria between the MTrPs group and the control group.** (A) Thirty photos were randomly selected from each group. The Image-Pro Plus 6.0 (Media Cybernetics, Inc., Rockville, MD, USA) image analysis system was used to select five complete sarcomeres using the 2.5 k 5 µm scale as the standard. Sarcomere length (µm) was measured. Additionally, five equal subregions were selected based on the 2.5 k 5 um scale for counting of mitochondria. (B) In the MTrPs group muscle fibres were broken (black arrow), mitochondrial cristae fracture deformation or cavitation occurred, and cytoplasmic oedema occurred (white arrow). The magnifications are × 2.5 k/7.0 k. The bars are 5.0/2.0 µm. *$p < 0.05$, ****$p < 0.0001$, compared to the control group, $n = 30$. The data were analysed by independent-sample T tests; repetitions = 3.

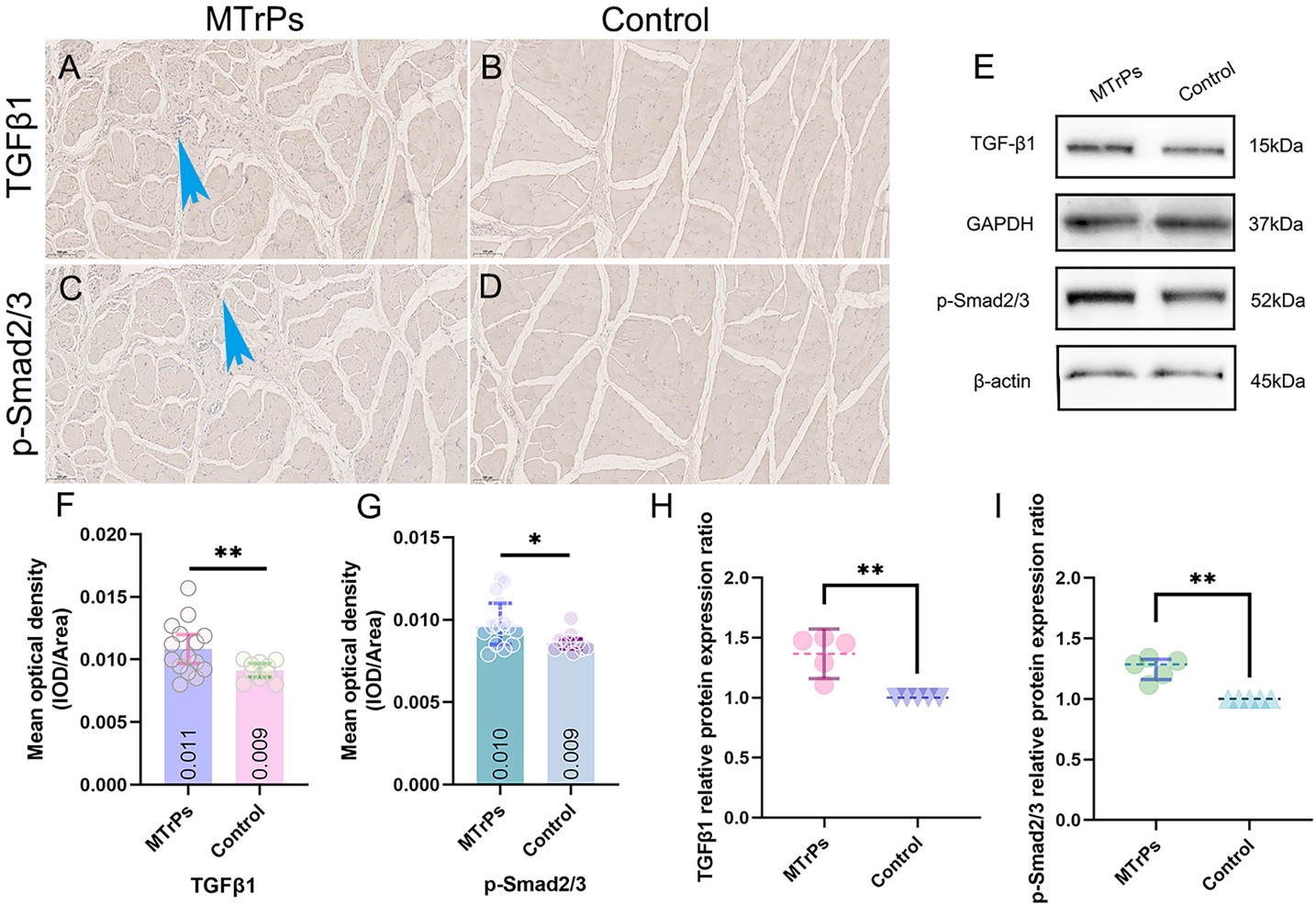

**Figure 5 The expression of TGF-β1 and p-Smad2/3 in the vastus medialis muscle of rats was determined by IHC and WB.** (A and C) More brown particles were observed around the disordered muscle fibres in the MTrPs group than in the control group (blue arrow). The control group showed smooth, continuous muscle fibres without contracture nodules (B and D). (E) Immunoblots showing TGF-β1 and p-Smad2/3 protein expression levels and (H and I) Western blot quantification relative to GAPDH/β-actin. $*p < 0.05$, $**p < 0.01$, compared to the control group. The data were analysed by independent-sample T tests or Mann−Whitney U tests; repetitions = 3.

2020; *Liang, Guo & Li, 2021*). Those studies confirmed that the stiffness of MTrPs tissue or the surrounding tissue is greater than that of normal tissue from the same location. In this study, we hypothesized that the TGF-β1/Smad2/3 signalling pathway is involved in skeletal muscle fibrosis during MTrPs formation.

Under normal circumstances, damaged skeletal muscle is degraded *via* inflammatory cell infiltration (*Mahdy, 2018*), and quiescent satellite cells are activated, proliferate, and differentiate to form new myotubes, which mature into myofibers, with concomitant production of new extracellular matrix (ECM), blood vessels, and nerves (*Laumonier & Menetrey, 2016*; *Yin, Price & Rudnicki, 2013*). However, the excessive accumulation of ECM components, especially the accumulation of collagen, *via* increased production, decreased degradation or both, can lead to the occurrence of skeletal muscle fibrosis (*Gillies & Lieber, 2011*). Excessive deposition of fibrous tissue can cause functional

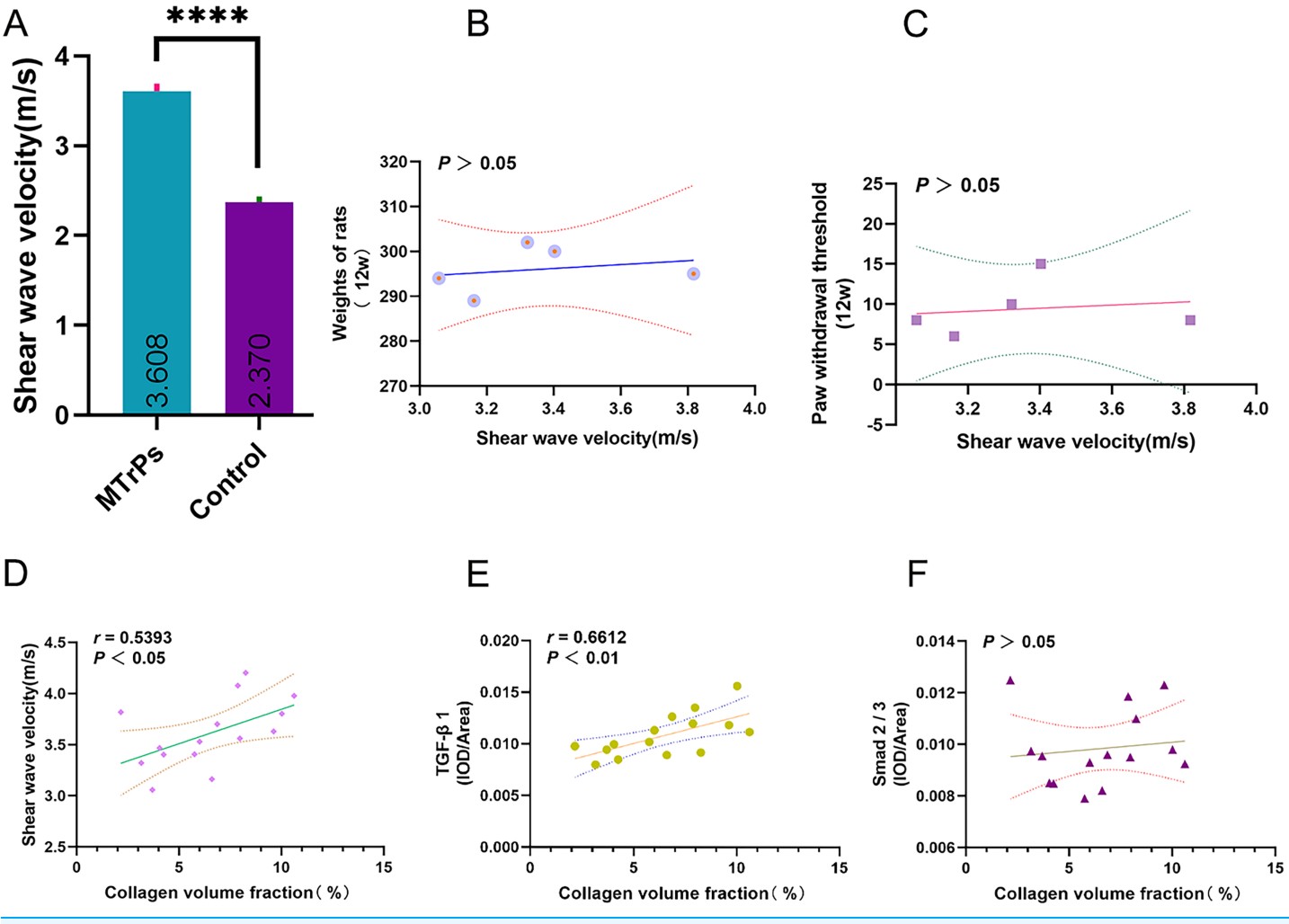

**Figure 6 Correlation plots.** ****p < 0.0001, compared to the Control group. The data were analysed by independent-sample T tests; repetitions = 3.

impairment of skeletal muscle and muscle fibre aplasia and increase the risk of muscle reinjury (*Järvinen et al., 2002*; *Prazeres et al., 2018*). The TGF-β family is a class of cytokines widely involved in cell growth, differentiation, apoptosis and tissue maturation (*Massagué, 2012*). The Smad protein family is a TGF-β signal transduction pathway in cells that is responsible for passing TGF-β signals from the cell membrane into the nucleus and then activating the transcription of downstream target genes (*Meng, Nikolic-Paterson & Lan, 2016*). However, certain pathological triggers, such as ischaemia–reperfusion injury, can lead to excessive TGF-β activation, thus causing excessive ECM deposition and progressive fibrosis (*Kim, Sheppard & Chapman, 2018*). TGF-β1 is also considered one of the most potent profibrotic factors and regulators of fibrosis development due to its control of ECM synthesis, remodelling and degradation (*Delaney et al., 2017*). During muscle repair, it is able to activate the expression of genes encoding ECM proteins. The TGF-β1/Smad signalling pathway is the main signalling pathway by which TGF-β1 plays its biological role. Smad2/3 is an extremely important protein in this signalling pathway.

Previous studies have also shown that the expression of TGF-β, Smad2/3, and p-Smad2/3 in the injured area of mouse skeletal muscle after acute blunt injury is increased and that p-Smad2/3 shows the most obvious increase (*Li et al., 2013*).

In this study, we found that TGF-β1 and p-Smad2/3 were more highly expressed in the MTrPs group than in the control group (Fig. 5). In the MTrPs group (Fig. 3), red muscle fibres were disordered and squeezed, collagen fibre deposition was increased significantly, and the fibres were thick and wrapped around new muscle cells and intermuscular membranes. Based on the above evidence, we conclude that MTrPs may exhibit fibrotic conditions. In the MTrPs group in the present study, the sarcomeres were shortened, many muscle fibres were broken, the number of mitochondria was reduced, and the mitochondrial cristae were fractured and deformed or cavitated. Studies have shown that inhibition of mitochondrial biosynthesis can lead to the occurrence of skeletal muscle fibrosis induced by collagen deposition (*Dulac et al., 2020*). In a model of age-related muscular atrophy, enhancement of mitochondrial enzyme activity and mitochondrial density was found to increase muscle mass and strength while inhibiting type I collagen deposition (*Leduc-Gaudet et al., 2019*). In skeletal muscle fibrosis induced by the braking model, application of the antioxidant astaxanthin can reduce the production of mitochondrial ROS in the muscle and attenuate skeletal muscle fibrosis (*Maezawa et al., 2017*). Therefore, mitochondrial function plays an important role in preventing skeletal muscle atrophy and skeletal muscle fibrosis.

In this study, we found that in the MTrPs group, the collagen volume fraction was linearly correlated with SWV and TGF-β1. The correlation coefficient between the collagen volume fraction and SWV was 0.5393 ($p < 0.05$). The correlation coefficient between the collagen volume fraction and TGF-β1 was 0.661 ($p < 0.01$) (Fig. 6). The above results verified our hypothesis that the TGF-β1 signalling pathway is involved in skeletal muscle fibrosis during the formation of MTrPs. Although the TGF-β/Smad pathway has long been considered to induce matrix synthesis and promote fibroblast contraction through fibroblasts, its specific mechanism of action in the context of MTrPs remains unclear. In the future, we will conduct in-depth studies on the TGF-β/Smad pathway at the gene level in combination with transcriptomics and proteomics analyses to further elucidate the pathogenesis of MTrPs. This study also provides guidance and theoretical support for clinical treatment.

## CONCLUSION

Fibrotic conditions may be involved in the formation of MTrPs. Ultrasound point shear wave elastography, TGF-β1 and p-Smad2/3 can reflect the degree of MTrPs fibrosis to some extent. Further exploration of the important role of TGF-β1-Smad2/3 in the pathogenesis of MTrPs will be of great significance for clinical treatment.

### Funding
The authors received no funding for this work.

## Competing Interests

The authors declare that they have no competing interests.

## Author Contributions

- Xin Fang conceived and designed the experiments, performed the experiments, analyzed the data, authored or reviewed drafts of the article, and approved the final draft.
- Yalong Yin performed the experiments, analyzed the data, prepared figures and/or tables, and approved the final draft.
- Haimei Lun performed the experiments, prepared figures and/or tables, and approved the final draft.
- Shitao Su performed the experiments, prepared figures and/or tables, and approved the final draft.
- Shangyong Zhu conceived and designed the experiments, authored or reviewed drafts of the article, and approved the final draft.

## Animal Ethics

The following information was supplied relating to ethical approvals (*i.e.*, approving body and any reference numbers):

This study was approved by the Animal Ethics Committee of Guangxi Medical University.

## Data Availability

The raw data is available in the Supplemental Files.

## Supplemental Information

Supplemental information for this article can be found online at http://dx.doi.org/10.7717/peerj.16588#supplemental-information.

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
