# Peer review of "Assessment of the effects of transforming growth factor beta1 (TGF-β1)-Smad2/3 on fibrosis in rat myofascial trigger points using point shear wave elastography"

_PeerJ, doi:10.7717/peerj.16588_

## Round 0.1 · original submission · Major Revisions

As you can see, the opinions of the reviewers about your paper are highly polarized. Although one of the reviewers recommended rejection, I decided to give you an opportunity to reply to critiques, address the pointed issues, and amend your manuscript accordingly.

**Language Note:** PeerJ staff have identified that the English language needs to be improved. When you prepare your next revision, please either (i) have a colleague who is proficient in English and familiar with the subject matter review your manuscript, or (ii) contact a professional editing service to review your manuscript. PeerJ can provide language editing services - you can contact us at copyediting@peerj.com for pricing (be sure to provide your manuscript number and title). – PeerJ Staff

Reviewer 1 ·

Basic reporting

no comment

Experimental design

Introduction:
“Athletes who are undergoing preseason training can suffer muscle overload injuries and develop trigger points as well.”
Why mention athletes here?

“Although both are noninvasive examination techniques, US has faster data acquisition and a lower cost than MRI. Point shear wave (STQ) elastography involves the use of a set of shear waves to induce normal and directional tissue displacement at a single focal point. The velocity of the shear waves vertical to the plane of induction is calculated and immediately transformed by Young’smodulus E to quantify stiffness(Nightingale 2011).”
Essentially, elastic ultrasonography is assessed based on "stiffness," a concept in material mechanics. However, is it truly precise enough to be a diagnostic indicator in diagnostics? In accordance with physiopathology, inflammatory lesions (not just limited to trigger points) can also become stiffer. After all, redness, swelling, heat, and pain are the fundamental manifestations of inflammation.

Materials & Methods:
(1)MTrP model rats:
"hypertension bands and contracture nodules were found on palpation,"
Is it necessary for the individual performing the palpation of the trigger points to possess a certain level of experience?
Can we consider the estimated duration of 8 weeks for the modeling process to be feasible and achievable?

(2)Ultrasound image processing:
“(L9-3 s) probe (5-14 MHz).”
Is there a disparity between the probe model and the labeled frequency range of detection? Specifically, can we consider a depth of detection of 14 MHz for the superficial musculature of the rat to be realistic?

"The ultrasonic probe was gently placed on the marks for scanning, and the MTrPs were observed in multiple sections to avoid anisotropy artefacts"
How to accurately determine the localization of trigger points under ultrasound? The precise positioning of the ultrasound probe in relation to the long axis of the muscle is not provided.

“After the EMG examination...”
How can we discern between areas of trigger points observed through ultrasound and localized injuries stemming from invasive electromyography testing?

“The data are expressed in m/s.”
Elastography is generally evaluated using Young's modulus in kPa, which is the shear wave velocity (SWV), why not use the recognized Young’smodulus E?

(3)Masson staining procedure:
In this article, only Masson staining is used to evaluate fibrosis, and collagen, fibroblast markers and other tests should also be supplemented.

(4)The biggest problem of trigger points is muscle dysfunction and chronic pain, and the article does not discuss the change of mechanical tenderness threshold and the correlation between existing indicators, and it is recommended to supplement and improve;

(5)Immunohistochemical assay:
In the TGF-β signaling pathway, serine (S) in the SXS motif (X refers to M or V) at the end of R-Smad C can be directly phosphorylated by type I receptors to lead to R-Smad activation, in which Smad2/3 is phosphorylated by type I receptors ALK4/5/7 of the TGF-β/activin/nodal subfamily, and the article only detects the normal Smad2/3, which is inappropriate.

Validity of the findings

Abstract:
"Background & Aims. To date, there have been few studies on the diagnostic value of point shear wave elastography (STQ) in a rat myofascial trigger point (MTrP) model based on transforming growth factor beta1 (TGFβ1)-Smad2/3 regulation of fibrosis."
To date, there have been few studies on the diagnostic value of point-shear wave elastography (STQ) in rat myofascial trigger point (MTrP) models based on transforming growth factor β 1 (TGF³1)-Smad 2/3 regulation. It does not provide a good illustration of the background and purpose of this article.

Result:
(1)The authors' organization of the EMG data is superficial, as they only provide qualitative presentations. A more comprehensive analysis could be conducted by examining waveform occupancy and amplitude frequency, and using this information for correlation analysis. After all, EMG can detect muscle dysfunction, which is a significant aspect of the provocative pain point.

(2)HE showed inconsistent muscle profiles, AC for the model group, bias for cross-section, and BD for blank group but bias for longitudinal section.

(3)“More brown particles are observed around the disordered muscle fibres in the MTrP group (A and C)”
However, the immunohistochemical images of TGF-β1 and Smad2/3 show no obvious difference in expression between the groups. Please add a high-resolution image of more than 200 times. In addition, please add WB experiments for quantitative analysis.

(4)Figure 7 shows us the collagen volume fraction (%) and shear wave velocity (m/s), collagen volume fraction (%) and TGFβ1, collagen volume fraction (%) and SMAD 2/3 expression in the MTrP group The linear correlation of TGFβ1 and collagen expression has been shown in quite a lot of strong research evidences.

Conclusions:
The conclusion should primarily focus on succinctly summarizing the key findings and implications of the study, rather than solely presenting the significance of the results.
The study undertook an examination of TGFβ1, Smad2/3, and SWV in both the MTrP group and the control group. Its objective was to evaluate their correlation and ascertain their potential as indicators of MTrP fibrosis. However, the current conclusion, which states that "ultrasound STQ elastography, TGFβ1, and Smad2/3 could reflect the degree of MTrP fibrosis to some extent," is not entirely accurate. To establish a more precise conclusion, it is imperative to incorporate additional experiments, such as positive and negative response experiments, as well as integrate a mechanical pain threshold test for further investigation.

Additional comments

Several studies have been published regarding the correlation betweenTGF-β 1 and MTrP, MTrP rat model of shear wave elastic modulus, fibrosis and MTrP. This paper further examines the myoelectricity, HE, Masson, electron microscopy, and other indicators of the MTrP model. However, it is worth noting that the current study does not introduce significant innovation when compared to the existing body of literature.
For example:
[1]王兴瑜.针刺激痛点联合电冲击波对肌筋膜激痛点MTrPs大鼠海马表达的影响[J].中医学报,2021,36(03):576-580.DOI:10.16368/j.issn.1674-8999.2021.03.123.
[2]景亚军,陈美雄,曹磊等.温和灸对肌筋膜激痛点TGF-β1/Smad4表达的影响[J].中华中医药学刊,2018,36(12):3019-3022+3110-3112.DOI:10.13193/j.issn.1673-7717.2018.12.051.
[3]吕恒勇,李真,王月香等.慢性肌筋膜激痛点大鼠模型的剪切波弹性模量研究[J].中国临床解剖学杂志,2017,35(01):57-61.DOI:10.13418/j.issn.1001-165x.2017.01.012.

Reviewer 2 ·

Basic reporting

The figures are not so professional, such as the format of Fig2 and Fig4, and the bar of Fig5 and Fig6 are not illustrated.

Experimental design

Can all the rats develop the MTrP model through a blunt strike in combination with eccentric exercise for 8 weeks? Are they all comply with the critria which described in the article?

Validity of the findings

no comment

---

## Round 0.2 · Minor Revisions

Please address the remaining concerns of reviewer #2 and amend Figure 1C accordingly.

Reviewer 1 ·

Basic reporting

no comment

Experimental design

no comment

Validity of the findings

no comment

Additional comments

no comment

Reviewer 2 ·

Basic reporting

no comment

Experimental design

no comment

Validity of the findings

The author showed the paw withdrawal threshold in Fig1C, but the bar of it was not illustrated.

---

## Round 0.3 · accepted · Accept

Thank you for addressing the remaining issues and for amending the manuscript accordingly. I am pleased to inform you that the revised manuscript is acceptable now.